# The Dilemma of TriHard Loss and an Element-Weighted TriHard Loss for Person Re-Identification

Yihao Lv, Youzhi Gu and Xinggao Liu

State Key Laboratory of Industry Control Technology
College of Control Science and Engineering
Zhejiang University
Hangzhou 310027, P.R. China
{lvyihao,gu_youzhi,lxg}@zju.edu.cn

## Abstract

Triplet loss with batch hard mining (TriHard loss) is an important variation of triplet loss inspired by the idea that hard triplets improve the performance of metric leaning networks. However, there is a dilemma in the training process. The hard negative samples contain various quite similar characteristics compared with anchors and positive samples in a batch. Features of these characteristics should be clustered between anchors and positive samples while are also utilized to repel between anchors and hard negative samples. It is harmful for learning mutual features within classes. Several methods to alleviate the dilemma are designed and tested. In the meanwhile, an element-weighted TriHard loss is emphatically proposed to enlarge the distance between partial elements of feature vectors selectively which represent the different characteristics between anchors and hard negative samples. Extensive evaluations are conducted on Market1501 and MSMT17 datasets and the results achieve state-of-the-art on public baselines. The implementation of this work is available at https://github.com/LvWilliam/EWTH-Loss.

## 1 Introduction

Person re-identification (ReID) is an important branch of computer vision. Many successful designs of classification networks [1][2][3] are applied as backbones of ReID networks to extract global features of samples [4][5][6][7]. It is efficient to train such networks as a multi-class classification task taking ids or other attributes as labels [8][9][10], which enhances the capability of networks to obtain more specific features. Some works focus on learning local features of samples, such as separated blocks of images [11][12], different semantical key points of bodies [13][14][15]. When more information is provided, such as ids of cameras [16][17] or viewpoints [18][19], networks can be trained to adapt the differences of inputs from multiple domains, which will improve the portability among different datasets. Other works concentrate on metric learning methods of ReID which are widely used in images retrieval area [20][21][22]. The key point is to learn distances between pairs of similar or dissimilar inputs [23]. Loss functions are essential for metric learning to realize the effect of metrics. Many works combine contrastive loss with softmax cross-entropy loss [11][24][25] to improve the performance with the advantage of metric and feature representation learning. Triplet loss was first proposed in FaceNet [26] and has applied in many works [27][28][29]. It's a preferable function to make the distances of anchors and positive samples closer than anchors and negative samples by a margin. FaceNet [26] also proposes an idea to calculate triplet loss by hard samples to extract more meaningful and intrinsic features, which has been studied widely [30][31][32]. Triplet loss with batch

hard mining (TriHard loss) was proposed in [31] which improves the way of generating hard triplets during training. It is applied as the metric loss in many SOTA baselines of ReID[26][33][34].

However, there is a dilemma in TriHard loss during training, which will hamper the result of clustering within classes. That is proved both theoretically and experimentally in this paper. And three ways to alleviate the dilemma are proposed:

1) Half TriHard loss. To transform the Euclidean distances between anchors and hard negative samples into a constant within a batch in TriHard loss.

2) TriHard loss with feature normalization. To make the gradients more stable during training.

3) Half TriHard loss with average negative samples. To reduce the obstruction of hard negative samples.

In this paper, we emphatically propose an element-weighted TriHard loss adapted from TriHard loss to eliminate the dilemma as much as possible. The weights are element-wise to output feature vectors of network:

$$T_{a,n_h}^i = f(|W_a^i - W_{n_h}^i|)(i \in [1, q]) \tag{1}$$

where $q$ is the dimension of output feature vectors, $a$ is the anchors and $n_h$ is the hard negative samples of anchors. $T_{a,n_h}^i$ is the weight of $i$ th element in feature vector. $W_a^i$, $W_{n_h}^i$ are the $i$ th elements of weight vectors corresponding to the ids of $a$ and $n_h$ in fully connected layers of classifier in ReID networks.

The frameworks of networks are not influenced that makes EWTH loss convinient to be used in existing metric learning methods of ReID. All the proposed losses are evaluated on Market 1501 [35] and MSMT17 [36] datasets, which shows remarkable improvements towards TriHard loss. The baselines in this paper are published works named reid-strong-baseline [37] and AGW [33].

## 2 Related works

Contrastive loss [38][24] is proposed based on the idea that person re-identification is a combination of feature representation and metric relation. Therefore, it is designed to enlarge or narrow the distances between samples with the different or same ids to force the clustering within classes. Triplet loss [26] contains an extra relation between the distances of positive and negative sample pairs compared with contrastive loss. That adds more freedom to the function so as to achieve better performance. Triplet center loss [39]combines triplet loss [26] and center loss [40] for object retrieval. It helps the features cluster to their centers of classes and enlarge the distance between samples and centers of other classes. Adaptive weighted triplet loss [41] is proposed to give different weights to different triplets. The weights are obtained by difficulty levels of triplets reflexed by the Euclidean distance. That makes the network focus on harder triplets. Quadruplet loss [42] improves triplet loss by raising the threshold of clustering within classes.The additional term forces the distances of positive pairs to be closer than random negative pairs in training dataset. Quadruplet loss has a severer condition than triplet loss which improves the performance. Triplet loss with batch hard mining (TriHard) loss [31] is a variation of triplet loss which solves a practical problem that the quantity of hardest triplets in the whole training set is much fewer than the remaining samples. Therefore, TriHard loss selects the hardest samples of each anchor online within batches instead of the whole training set. Margin sample mining loss (MSML) [43] combines quadruplet loss and TriHard loss. It selects a hardest triplet in a batch. It has a harder condition than TriHard loss.

## 3 Formulation

### 3.1 Triplet loss and TriHard loss

Triplet loss [26] makes Euclidean distances between feature vectors from different classes larger than that from the same class by a constant :

$$L_T = \sum_{a,p,n \in D} [d(a, p) - d(a, n) + \alpha]_+ \, (\alpha > 0) \tag{2}$$

where $a, p, n \in R^q$ are feature vectors of anchors, positive and negative samples. $d(x,y)$ is the Euclidean distance between $x$ and $y$. $D$ can be the whole training dataset or mini-batch during training.

TriHard loss [31] adapts the triplet selection method to obviously compress the size of mini-batch compared with [26] to make the training of large datasets practical:

$$L_{TH} = \sum_{a, p_h, n_h} [d(a, p_h) - d(a, n_h) + \alpha]_+ \ (\alpha > 0) \tag{3}$$

where $a$ are all the vectors of samples in the mini-batch $D$, $p_h$ and $n_h$ are vectors of the hardest positive and negative samples towards anchors within $D$. $D$ contains $P \times K$ samples which is composed of $P$ randomly selected ids of persons in training data and $K$ samples of each id. That makes the hardest triplets more soft and various, which avoids overfitting on a few hardest triplets of the whole dataset. It solved the problem mentioned in [26] that the hardest samples on the whole dataset towards anchors are easy to cause the convergence to local minima.

## 3.2 Dilemma in TriHard loss

The core idea of TriHard loss [31] is sensible. However, it omits a key point that hard negative samples share certain characteristics in common with anchor samples and hard positive samples. The elements of feature vectors extracted from these characteristics are forced to be away from each other by $(a, n_h)$ pairs but close to each other by $(a, p_h)$ pairs. Fig.1 shows 3 hard triplets with the order $a - p_h - n_h$. The appearance of clothes are the most obvious characteristics of persons which cover most areas of pictures and will be reflected by some certain elements of the feature vectors. But hard negative samples also share quite similar appearance of clothes that hamper these elements to cluster. According to equation 3, take only one triplet $(a, p_h, n_h)$ as example, TriHard loss can be formulated as:

$$L_{TH} = [\|a - p_h\|_2 - \|a - n_h\|_2 + \alpha]_+ \ (\alpha > 0) \tag{4}$$

Calculate partial derivative of $L_{TH}$ with respect to the $O_{k,c,h,w}$ which is a cell of $F_l$ at the position $(k, c, h, w)$, where $F_l$ is the last layer of convolutional filters:

$$\frac{\partial L_{TH}}{\partial O_{k,c,h,w}} = \overbrace{\frac{a_k - p_h^k}{\|a - p_h\|_2} \left( \frac{\partial a_k}{\partial O_{k,c,h,w}} - \frac{\partial p_h^k}{\partial O_{k,c,h,w}} \right)}^{part1} - \overbrace{\frac{a_k - n_h^k}{\|a - n_h\|_2} \left( \frac{\partial a_k}{\partial O_{k,c,h,w}} - \frac{\partial n_h^k}{\partial O_{k,c,h,w}} \right)}^{part2} \tag{5}$$

where $x_k \in R$, $x$ is $a, p_h, n_h$ and $k \in [1, q]$. $x_k$ is the result of global average pooling on channel $k$ of the last layer of feature maps in the network. $O_{k,c,h,w}$ is related to the $k$th element of vector $a$, $p_h$ and $n_h$. It shows the influence of $p_h$ and $n_h$ are totally opposite towards equation 5. $\frac{\partial x_k}{\partial O_{k,c,h,w}} = \delta x_k^{'} \in R$ where $x_k^{'}$ is the result of global average pooling on channel $k$ of the second last layer of feature maps in the network. $\delta \in R$ is a constant. And each channel of feature maps represents a same class of characteristics of inputs in a network [44]. Therefore, when channel $k$ of the last and channel $c$ of the second last layer of feature maps are both sensitive toward similar characteristics, such as colors of shirts or jeans shown in Fig.1, $\frac{\partial x_k}{\partial O_{k,c,h,w}}$ will be close and $x_k$ will also be close in value. And $\|a - p_h\|_2$, $\|a - n_h\|_2$ are chosen to be near in value by TriHard loss on Euclidean distance. So, $part1$ and $part2$ will be close in value. That may cause instability of $\frac{\partial L_{TH}}{\partial O_{k,c,h,w}}$ which may fluctuate near zero during training. It's harmful for $O_{k,c,h,w}$ to learn features from these characteristics which

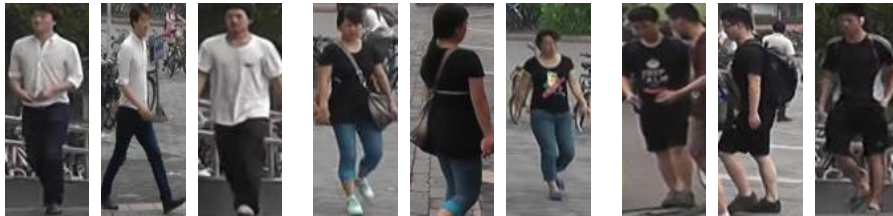

Figure 1: Anchors, the hardest positive and negative sample selections in mini-batch during training

obstructs the clustering within classes. That will also happen to other elements of feature vectors that are sensitive towards similar characteristics in hard triplets. According to chain derivative law, filters in lower layers will be influenced as well.

### 3.3 Several mitigation strategies of TriHard loss dilemma

#### 3.3.1 Half TriHard loss

To solve the problem formulated in equation 5, it is necessary to eliminate the obstruction of hard negative samples during features clustering within classes. TriHard loss can be adapted as:

$$L_{HTH} = \sum_{a,p_h,n_h \in D} [d(a, p_h) - \beta + \alpha]_+ \, (\alpha > 0) \tag{6}$$

where $\beta = d(a, n_h)$ is calculated online in each batch. Therefore, it is constant within batches but variable between batches. Take only one triplet $(a, p_h, n_h)$ as example, the partial derivative of $L_{HTH}$ with respect to $O_{k,c,h,w}$ is

$$\frac{\partial L_{HTH}}{\partial O_{k,c,h,w}} = \frac{a_k - p_h^k}{\|a - p_h\|_2} \left( \frac{\partial a_k}{\partial O_{k,c,h,w}} - \frac{\partial p_h^k}{\partial O_{k,c,h,w}} \right) \tag{7}$$

where the influence of hard negative samples is eliminated in gradient and the network can focus on learning features from mutual characteristics within classes. There is still a margin $\alpha$. Half TriHard loss improves the performance of clustering obviously compared with TriHard loss.

But $L_{HTH}$ abandons the term of repelling which may harm the performance of network when the distances among feature vectors of different clusters are originally close. That should be considered case by case.

#### 3.3.2 TriHard loss with feature normalization

Feature normalization (FN) is widely used in face recognition [45][46]:

$$\hat{F} = \frac{\gamma F}{\|F\|_2} (\gamma > 0) \tag{8}$$

where $F \in R^m$ is a feature vector and $\gamma$ is an amplitude parameter to control the norm of vectors. Equation 8 is often applied before softmax [45][46] operation in face recognition area. That makes the equivalence of Euclidean distance and cosine distance.

To utilize FN before calculating TriHard loss can also alleviate the dilemma. With FN ($\gamma = 1$) in equation 8, equation 5 can be transformed as:

$$\frac{\partial L_{TH}}{\partial O_{k,c,h,w}} = -k_1 \left[ \left( \frac{p_h^k}{\|a\|_2 \|p_h\|_2} - \frac{a_k \cos(a, p_h)}{\|a\|_2^2} \right) \frac{\partial a_k}{\partial O_{k,c,h,w}} + \left( \frac{a_k}{\|a\|_2 \|p_h\|_2} - \frac{p_h^k \cos(a, p_h)}{\|p_h\|_2^2} \right) \frac{\partial p_h^k}{\partial O_{k,c,h,w}} \right]$$

$$+ k_2 \left[ \left( \frac{n_h^k}{\|a\|_2 \|n_h\|_2} - \frac{a_k \cos(a, n_h)}{\|a\|_2^2} \right) \frac{\partial a_k}{\partial O_{k,c,h,w}} + \left( \frac{a_k}{\|a\|_2 \|n_h\|_2} - \frac{n_h^k \cos(a, n_h)}{\|n_h\|_2^2} \right) \frac{\partial n_h^k}{\partial O_{k,c,h,w}} \right] \tag{9}$$

where $k_1 = \frac{\sqrt{2}}{2} \frac{1}{\sqrt{1 - \cos(a, p_h)}}$, $k_2 = \frac{\sqrt{2}}{2} \frac{1}{\sqrt{1 - \cos(a, n_h)}}$

Even $p_h$ and $n_h$ are still the hardest ones before FN, they are selected by Euclidean distance which indicates $\cos(a, p_h)$ and $\cos(a, n_h)$ are uncertain. Therefore, terms of equation 9 are more random in value compared with equation 5, which makes the result more stable instead of fluctuating around zero before convergence during training. That alleviates the dilemma in Section 3.2. In the meanwhile, FN unifies the norm of vectors, which reduces the risk of overfitting and makes the network more robust on different domains of training and testing datasets.

According to equation 8, FN will limit all the ends of feature vectors onto the surface of a m-dimensional hypersphere. That will cause the distortion of relative Euclidean distance between $a$, $p$ and $n$. It's proved in supplementary material. Therefore, the performance of networks with FN may even poorer, which is also a case-by-case problem.

### 3.3.3  Half TriHard loss with average negative samples

According to equation 5, when channel $k$ of the last layer of feature maps is sensitive toward a kind of characteristics that exists in $a$ and $p_h$ but not in $n_h$, such as different colors of clothes or figures on shirts between different persons but the same among images of one person, $part2$ tends to be dominant during training which will make $k$th element of feature vectors repel others between $a$ and $n_h$. That extends the distances among different clusters. But in TriHard loss, samples of $n_h$ have more similar characteristics than common ones. That harms the clustering within classes more than repelling among different classes. Therefore, a compromise can be adopted:

$$L_{HNTH} = \overbrace{\sum_{a,p_h,n_h \in D} [d(a,p_h) - \beta_1 + \alpha_1]_+}^{part1} + \overbrace{\sum_{a,p_h,n \in D} [\beta_2 - \overline{d(a,n)} + \alpha_2]_+}^{part2} (\alpha_1, \alpha_2 > 0) \quad (10)$$

where $\beta_1 = d(a,n_h)$, $\beta_2 = d(a,p_h)$, $\overline{d(a,n)} = \frac{1}{N}\sum_{a,n \in D} d(a,n)$. $Part1$ is half TriHard loss that focuses on the clustering within classes and $part2$ emphasizes repelling between different classes. $Part2$ of equation 10 reduces the influence of $n_h$ on clustering within classes in a batch and retains the function of repelling features away among different classes.

But the dilemma of TriHard loss [31] will be obviously residual when quite a few negative samples are hard to anchors. In this case, the dilemma of TriHard loss are still severe. And the average negative samples are intrinsically weaker to enlarge the gaps between anchors and negative samples compared with the hardest negative samples in a batch. Therefore, $L_{HNTH}$ may perform poorer which is also a case-by-case problem.

## 4  Element-weighted TriHard loss

### 4.1  Why elements of feature vectors need to be weighted

Fig.2 shows attention levels of a ReID network on several images, which are obtained through the weights in the classifier of the network and the last layer of feature maps [44]. The network tends to focus on most parts of persons in images. That is effecient for common samples which account for the majority of training and testing sets. But for hard samples, it is not enough. In Fig.1, those differences between samples of anchors and hardest negative samples cover only small areas of images, which are used to tell 2 persons apart. But a single network cannot pay much attention on these tiny areas which will decrease the diversity of feature vectors, so that the performance on common samples will turn worse. TriHard loss aims to improve the capacity of networks on hard samples but faces a dilemma discussed in this paper. To solve the problem, an important idea is proposed that vectors of anchors and hardest negative samples could also have similar elements. Only elements in features sensitive to different characteristics between samples of $a$ and $n_h$ are trained to repel. The mistake that TriHard loss makes is to repel the whole vectors between $a$ and $n_h$ ignoring the similarity in the original images and also feature vectors. That leads to the dilemma mentioned above. To realize this idea, elements of feature vectors are weighted before calculating Euclidean distance. But to locate the different areas in samples of $a$, $n_h$ and find which elements of feature vectors reflect them is a hard work. We utilize a simple method with the help of classification part in ReID network.

### 4.2  Element-weighted TriHard loss

In networks of person re-identification, triplet and classification loss are usually employed together. The ids of input samples are labels. $W_c \in R^{g \times q}$ is the weights of fully connected layer, where $q$ is the dimension of feature vectors and $g$ is the number of classes in training set. $r_{g_i} = W_c^{g_i,all} v$ is the score of vector $v$ belonging to class $g_i$ where $W_c^{g_i,all} \in R^{1 \times q}$ and $v \in R^q$. And sgn($W_c^{g_i,q_i}$) indicates the positive or negative belief of characteristic reflected by element $q_i$ of features belonging to class $g_i$ [44]. Equation 11 indicates the degree of difference between 2 classes of samples on each element of feature vectors.

$$W_{g_1,g_2} = \left| W_c^{g_1,all} - W_c^{g_2,all} \right| \quad (11)$$

where $g_1$, $g_2$ are the labels of samples1 and sample2, $|\cdot|$ is absolute value. When $g_1$, $g_2$ are the ids of $a$ and $n_h$ in a batch, we obtain the degree of difference in feature vectors of them. That is visualized in Fig.3 by method of [44] with $W_{g_1,g_2}$. The uniqueness compared with the other image is shown.

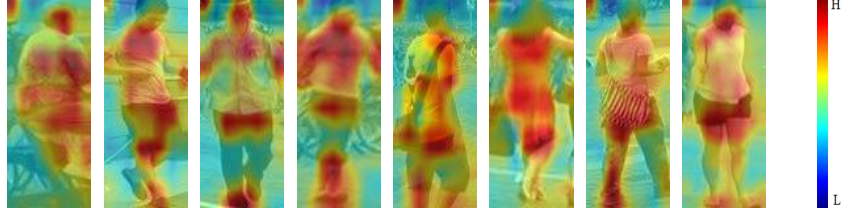

Figure 2: The attention level of network visualized on the input samples

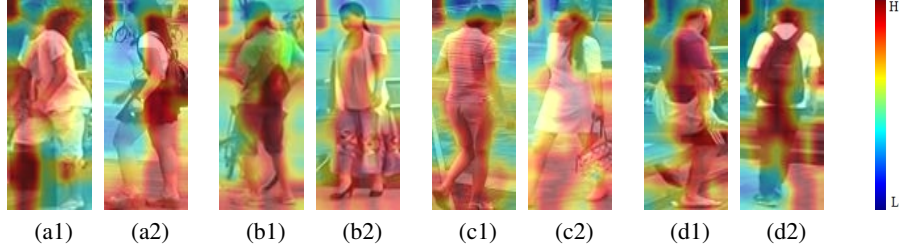

| (a1) | (a2) | (b1) | (b2) | (c1) | (c2) | (d1) | (d2) |

Figure 3: The uniqueness level visualized on the pairs of input samples

For example, in image (a1), his shoulder and the stuff he carries, in image (a2), the bag, her skirt and hair, are most unique to the other image. Therefore, it is meaningful to roughly locate the different areas in a pair of samples though the weights of classifier in the network.

The weights of EWTH is formulated as:

$$T_{a,n_h}^k = \frac{W_{g_a,g_{n_h}}^k}{max(W_{g_a,g_{n_h}})} + b \ (if \ \frac{W_{g_a,g_{n_h}}^k}{max(W_{g_a,g_{n_h}})} \geq t) \ or \ T_{a,n_h}^k = 0 \ (if \ \frac{W_{g_a,g_{n_h}}^k}{max(W_{g_a,g_{n_h}})} < t) \quad (12)$$

where $k \in [1,q]$ are the indexes of elements in weight vectors of EWTH loss and feature vectors. $b$ is a learnable parameter, $t \in [0,1]$ is a constant. EWTH loss is formulated as:

$$L_{EWTH} = \overbrace{L_{HTH}}^{part1} + \overbrace{\sum_{a,p_h,n_h \in D} [d(T_{a,n_h} \times a, T_{a,n_h} \times p_h) - d(T_{a,n_h} \times a, T_{a,n_h} \times n_h) + \alpha]_+}^{part2} \quad (13)$$

where $\times$ is element-wise multiplication and $(\alpha > 0)$. Take only one triplet $(a,p_h,n_h)$ as example, the partial derivative of EWTH loss with respect to $O_{k,c,h,w}$:

$$\begin{aligned}
\frac{\partial L_{EWTH}}{\partial O_{k,c,h,w}} =& [m_1 + (T_{a,n_h}^k)^2] \frac{a_k - p_h^k}{m_1 \|a - p_h\|_2} \left( \frac{\partial a_k}{\partial O_{k,c,h,w}} - \frac{\partial p_h^k}{\partial O_{k,c,h,w}} \right) \\
&- (T_{a,n_h}^k)^2 \frac{a_k - n_h^k}{m_2 \|a - n_h\|_2} \left( \frac{\partial a_k}{\partial O_{k,c,h,w}} - \frac{\partial n_h^k}{\partial O_{k,c,h,w}} \right)
\end{aligned} \quad (14)$$

where $k \in [1,q]$ and $m_1 = \|a \times T_{a,n_h} - p_h \times T_{a,n_h}\|_2 / \|a - p_h\|_2$, $m_2 = \|a \times T_{a,n_h} - n_h \times T_{a,n_h}\|_2 / \|a - n_h\|_2$. When the $k$th element of feature vectors reflects similar characteristics between $a$ and $n_h$, $T_{a,n_h}^k$ is very likely to be zero to facilitate the clustering of $a_k$ and $p_h^k$. On the other hand, $T_{a,n_h}^k$ will be positive to assist repelling $a_k$ and $n_h^k$.

When combined with $L_{HNTH}$, it is formulated as:

$$L_{NEWTH} = L_{EWTH} + L_{HNTH}^{p2} \quad (15)$$

Where $L_{HNTH}^{p2}$ is $part2$ of $L_{HNTH}$. That helps the network to enlarge the distances among different clusters better.

## 5  Experiment results

The baselines in this paper are reid-strong-baseline (bags of tricks) [37] and AGW [33]. We utilize ResNet50 [47] as the backbone. Tricks in reid-strong-baseline are warmup, random erasing aug-

Table 1: The results of ablation study of proposed losses on testing set

| Method | Market1501 | | MSMT17 | | Method | Market1501 | | MSMT17 | |
|--------|------------|--------|--------|--------|--------|------------|--------|--------|--------|
| | mAP | rank-1 | mAP | rank-1 | | mAP | rank-1 | mAP | rank-1 |
| B+TH | 85.6% | 94.1% | 45.1% | 63.9% | A+TH | 87.7% | 95.0% | 48.4% | 67.9% |
| B+HTH | 86.6% | 94.6% | 45.0% | 63.9% | A+HTH | 88.1% | 95.4% | 48.1% | 67.6% |
| B+HNTH | 87.3% | 94.9% | 44.6% | 63.8% | A+HNTH | 88.1% | 95.6% | 47.2% | 66.4% |
| B+EWTH | 87.7% | 95.0% | 48.7% | 67.8% | A+EWTH | 88.5% | 95.4% | 50.4% | 69.6% |
| B+NEWTH | **88.4%** | **95.1%** | **49.7%** | **68.1%** | A+NEWTH | **89.4%** | **95.6%** | **53.1%** | **71.5%** |
| B+TH+FN | 86.3% | 94.1% | 45.2% | 63.8% | A+TH+FN | 88.0% | 95.1% | 47.7% | 66.3% |

Table 2: The the relationships among feature vectors of anchors, positive and negative samples on training set

| Method | Market1501 | | | MSMT17 | | |
|--------|-----------|-----------|--------------|-----------|-----------|--------------|
| | $d_{ap}$ | $d_{an}$ | $d_{ratio}$ | $d_{ap}$ | $d_{an}$ | $d_{ratio}$ |
| B+TH | 12.915 | 27.922 | 2.162 | 16.830 | 32.363 | 1.923 |
| B+HTH | 10.150 | 23.110 | 2.277 | 14.616 | 27.978 | 1.914 |
| B+HNTH | 10.389 | 23.935 | 2.304 | 15.298 | 28.910 | 1.890 |
| B+EWTH | 10.022 | 25.181 | 2.513 | 13.669 | 29.473 | 2.156 |
| B+NEWTH | 10.249 | 25.828 | 2.520 | 13.806 | 29.365 | 2.127 |
| B+TH+FN | 12.624 | 28.508 | 2.258 | 15.989 | 32.150 | 2.011 |
| A+TH | 13.937 | 28.671 | 2.057 | 17.949 | 34.565 | 1.926 |
| A+HTH | 9.492 | 22.947 | 2.418 | 14.333 | 27.583 | 1.924 |
| A+HNTH | 9.610 | 23.671 | 2.463 | 14.874 | 28.367 | 1.907 |
| A+EWTH | 9.383 | 25.310 | 2.697 | 13.643 | 29.852 | 2.188 |
| A+NEWTH | 9.553 | 25.910 | 2.712 | 13.108 | 28.830 | 2.199 |
| A+TH+FN | 13.723 | 28.947 | 2.109 | 17.392 | 33.291 | 1.914 |

mentation [48], label smoothing [49], one last stride and BNNeck [37]. AGW contains all tricks. All the training strategies are default in both baselines. Comparative experiments are evaluated on dataset Market1501[35] and MSMT17 [36]. The default metric loss functions of bags of tricks (BoT) and AGW are TriHard loss [31] and weighted regularization triplet (WRT) loss [33]. And the classification loss is softmax cross-entropy loss. B + * and A + * denote the metric loss of BoT and AGW are replaced by $L_*$ in Table 1. And there is a hyper-parameter $t$ in $L_{EWTH}$ and $L_{NEWTH}$ and different values of $t$ are tested, which are listed in supplementary materials. The $\gamma = 1$ in FN and initial value of $b$ is 1. Table 1 shows the best results of all tested $t$ in each method. During training, element-weights of EWTH series losses are calculated online by equation 11 and 12. The gradients of $W_c$ in equation 11 is cut off to avoid having an effect on the training of classification module.

## 5.1 Ablation study

Table 1 lists the results of ablation study of proposed losses. Table 2 shows the average distances between anchors and positive samples ($d_{ap}$), anchors and negative samples ($d_{an}$) on training dataset after training. In the meanwhile, the distance ratios ($d_{ratio}$) is defined as equation 16

$$d_{ratio} = \frac{d_{an}}{d_{ap}} \qquad (16)$$

which evaluates the relationships among feature vectors of anchors, positive and negative samples in the training process. High values denote lower $d_{ap}$ which is better clustering within classes and higher $d_{an}$ which is wider gaps among different classes. It reflects a pattern between training and testing results of different losses in Table 1 and Table 2. That is the loss with higher value of $d_{ratio}$ on training dataset tends to perform better on testing set.

It is obvious that dataset MSMT17 is harder than Market1501 with lower mAP, rank-1 and $d_{ratio}$. The $d_{ap}$ of $L_{HTH}$ is lower than $L_{TH}$ which makes the clustering within classes better when the dilemma of TriHard loss is mitigated. $L_{HTH}$ performs better than $L_{TH}$ on Market1501 testing set and produces higher $d_{ratio}$ on training set. That is the clustering within classes is better and the gaps

Table 3: The results of different losses implemented on AGW baseline

| Losses | Market1501 | | Losses | MSMT17 | |
| --- | --- | --- | --- | --- | --- |
| | mAP | rank-1 | | mAP | rank-1 |
| MSML [43] | 78.4% | 91.6% | MSML[43] | 39.2% | 59.8% |
| ATL*[50] | 85.9% | 94.7% | ATL[50] | 42.2% | 62.4% |
| TML[51] | 86.4% | 94.3% | AWTL*[41] | 43.9% | 64.3% |
| AWTL*[41] | 86.8% | 95.3% | TML[51] | 45.1% | 64.4% |
| QLL [42] | 87.0% | 94.5% | OARTL*[52] | 45.3% | 65.3% |
| HERTL*[53] | 87.3% | 95.2% | HERTL*[53] | 45.9% | 65.3% |
| OARTL*[52] | 87.4% | 95.0% | WQLL*[54] | 46.5% | 65.3% |
| WQLL* [54] | 87.7% | 95.0% | QLL[42] | 47.1% | 66.0% |
| THL [31] | 87.7% | 95.0% | WSMTL*[54] | 48.2% | 66.6% |
| TFL[55] | 87.9% | 95.3% | SMTL[31] | 48.3% | 66.9% |
| SMTL[31] | 88.2% | 95.2% | THL [31] | 48.4% | 67.9% |
| WRTL* [33] | 88.2% | 95.2% | TFL[55] | 48.9% | 67.5% |
| WSMTL* [54] | 88.4% | 95.2% | WRTL*[33] | 49.5% | 68.2% |
| EWTH* (our) | 88.5% | 95.4% | EWTH* (our) | 50.4% | 69.6% |
| NEWTH* (our) | **89.4%** | **95.6%** | NEWTH* (our) | **53.1%** | **71.5%** |

between anchors and negative samples are still wide enough. But $L_{HTH}$ performs poorer than $L_{TH}$ on MSMT17 testing set and produces lower $d_{ratio}$ on training set. That indicates the negative samples tend to be more similar towards anchors with not large enough $d_{an}$ compared with $d_{ap}$. It reflects the intrinsic difference between 2 datasets and the problem $L_{HTH}$ may cause. Therefore, the repelling term is needed. In Table 2, $d_{ap}$ and $d_{an}$ of $L_{HNTH}$ are larger than $L_{HTH}$ but lower than $L_{TH}$. The condition of average negative samples are weaker than $L_{TH}$ which mitigates the dilemma but still affects the clustering within classes. $L_{HNTH}$ performs better than $L_{HTH}$ on Market1501 with higher $d_{ratio}$ but poorer on MSMT17 with lower $d_{ratio}$. That agrees with the comparison of $L_{TH}$ and $L_{HTH}$. On MSMT17, $d_{ratio}$ becomes lower from $L_{TH}$ to $L_{HTH}$ which indicates $L_{HTH}$ will result in lower $d_{an}$ compared with $d_{ap}$. That will amplify the residual dilemma of TriHard loss in $L_{HNTH}$. Therefore, the effect of $L_{HTH}$ and $L_{HNTH}$ varies among different datasets and even baselines. $L_{EWTH}$ replaces $part2$ of equation 10 with $part2$ of equation 13. The overall performance is much better than $L_{HNTH}$. The $d_{an}$ is larger and $d_{ap}$ is smaller. It eliminates the fights between clustering and repelling terms in the loss. $L_{NEWTH}$ produces the best results among all losses. The increments are 2.8%, 1.0% and 4.6%, 4.2% on BoT [37], 1.7%, 0.6% and 4.7%, 3.6% on AGW [33] about mAP and rank-1 on 2 datasets. In Table 1, to observe the comparison of results between $L_{HTH}$, $L_{HNTH}$ and $L_{EWTH}$, $L_{NEWTH}$ on dataset MSMT17, the average negative repelling results in the opposite effect. In $L_{NEWTH}$, those feature vectors of hardest negative samples are farther from those of anchors with the help of $L_{EWTH}$. Therefore, when calculating $L_{HNTH}^{p2}$, the influnce of these hardest negative samples are weaker which alleviates the dilemma in average negative samples and enlarge the gaps between anchors and common negative samples. The values of $d_{ap}$ of $L_{TH}$ with FN are little smaller than those of $L_{TH}$ which shows better clustering within classes. That is what we have proved in Chapter 3.3.2. In Table 1, $L_{TH}$ with FN performs better on Market1501 but poorer on MSMT17. The reason can be the range of feature vectors. In Table 2, the distribution range of feature vectors of MSMT17 are wider, so FN will cause more serious distortion proved in the supplementary materials. That will harm the accuracy of metric loss during training to produce poorer results.

## 5.2 Comparison with different losses and SOTA results

There are 13 variations of triplet loss [26] tested on AGW [33] baseline on Market1501 [35] and MSMT17 [36]. Losses with * improve triplet loss by weights. EWTH loss and NEWTH loss perform top1 and top2 in Table 1 that are made comparison with other losses in Table 3. EWTH and NEWTH loss achieve 2 highest results of mAP and rank-1 on Market1501 and MSMT17. Other weighted loss functions in Table 3 focus on calculating the importance and difficulty of each selected triplet though Euclidean distance [33][50][41][53], average precision (AP) [52] or additional weights [54]. These works consider relations among different triplets by giving more importance to harder samples. Element-weighted TriHard loss proposed in this paper concentrates on relations within elements of

Table 4: The comparison with state-of-the-art results of baselines

| Method | Market1501 | | MSMT17 | |
|---|---|---|---|---|
| | mAP | rank-1 | mAP | rank-1 |
| BoT [37] | 85.6% | 94.1% | 45.1% | 63.9% |
| AGW [33] | 88.2% | 95.2% | 49.5% | 68.2% |
| NEWTH (our) on BoT | 88.4% | 95.1% | 49.7% | 68.1% |
| NEWTH (our) on AGW | **89.4%** | **95.6%** | **53.1%** | **71.5%** |

feature vectors rather than relations between features. That solves the contradiction embedding in the training of triplet loss with hard samples though semantic information learned by the network. That performs better than those less meaningful information such as Euclidean or cosine distance. The remaining loss functions improve selection strategies of triplets [31][51], metric conditions [43][42][55] or abandon the margin[31].

Reid-strong-baseline (bags of tricks) [37], AGW [33] are two public state-of-the-art baselines of person re-identification. AGW is designed above bags of tricks (BoT) with 3 extra trick: non-local Attention Block [56], Generalized-mean (GeM) Pooling [57] and weighted Regularization Triplet (WRT) loss [33] which achieves better results compared with BoT. The results of original baselines and improved versions by NEWTH loss are rearranged in Table 4. Apparently, NEWTH loss on AGW outperforms these two state-of-the-art baselines and achieve SOTA above them. That proves the proposed element-weighted TriHard loss is more efficient.

# 6 Conclusion

In this paper, the dilemma of TriHard loss [31] is proved and several mitigation strategies are proposed. In the meanwhile, an element-weighted TriHard loss is designed and combined with the mitigation strategies. All the losses are evaluated on Market1501 [35] and MSMT17 [36] datasets and achieve state-of-the-art results on public baselines BoT [37] and AGW [33].

# Broader Impact

The shortcoming of TriHard loss [31] is proved theoretically and a series of element-weighted TriHard losses is proposed and tested in this paper. It is strongly explainable and easy to implement, which can be used in the existing methods of person re-identification (ReID).

The most obvious positive outcome is that improving the accuracy of results makes the automatic person identification technology more practicable in security, autonomous driving and other fields. That will improve efficiency and effectiveness of work or save human costs in these areas. And the ideas proposed in this paper may inspire more valuable and innovative researches in the future.

The negative outcomes result from the increasing accuracy of person identification and more used surveillance cameras. That may raise the risk of privacy breaches and other security issues, which may put everyone under monitoring. In the meanwhile, The public datasets for person ReID are usually non-consensual surveillance data, which are supposed to be an invasion of privacy. Therefore, the collection process should be public and informed to everyone who is contained in the collection. And the utilization of these datasets should be is subject to scrutiny and regulation.

# Acknowledgments and Disclosure of Funding

This work is supported by National Key R&D Program of China (grant number 2018YFB2004200), Zhejiang Provincial Natural Science Foundation (grant number LY18D060002), National Natural Science Foundation of China (grant numbers 61590921), and their supports are thereby acknowledged.

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
