[Supplementary Material]

# The Dilemma of TriHard Loss and an Element-Weighted TriHard Loss for Person Re-Identification

Yihao Lv, Youzhi Gu and Xinggao Liu

State Key Laboratory of Industry Control Technology
College of Control Science and Engineering
Zhejiang University
Hangzhou 310027, P.R. China
{lvyihao,gu_youzhi,lxg}@zju.edu.cn

## Supplementary materials

### A. Distortion of feature normalization on Euclidean distance

As is shown in Fig.1, we take 2-dimensioal space as an example. $\vec{a}$, $\vec{p}$, $\vec{n}$ and $\vec{a'}$, $\vec{p'}$, $\vec{n'}$ are origin and normalized feature vectors of anchors, positive and negative samples. Therefore, $|\vec{a} - \vec{p}| \in [|\vec{a}|\sin\theta_1, +\infty)$, $|\vec{a} - \vec{n}| \in [|\vec{a}|\sin\theta_2, +\infty)$ while $\left|\vec{a'} - \vec{p'}\right| = 2\gamma\sin\frac{\theta_1}{2}$, $\left|\vec{a'} - \vec{n'}\right| = 2\gamma\sin\frac{\theta_2}{2}$ ($\theta_1,\theta_2 \in [0,\pi]$). The relationship between $\left|\vec{a'} - \vec{p'}\right|$ and $\left|\vec{a'} - \vec{n'}\right|$ is determined by $\theta_1$ and $\theta_2$ but the relationship between $|\vec{a} - \vec{p}|$ and $|\vec{a} - \vec{n}|$ is unsure in the original space. Feature normalization distorts the real relative Euclidean distance between feature vectors which is very important in triplet loss of metric learning.

Figure 1: Graphical representation of feature normalization

### B. The results of series of EWTH losses with different value of $t$

Nulls in Table 1, 2, 3, 4 indicates the experiments under those values of $t$ are not conducted. Therefore, there are 5 different values of $t$ tested of each loss.

Table 1: Results of series of EWTH loss in BoT with different values of $t$ on Market1501

| Method | Values of $t$ | | | | | | | | | |
|---|---|---|---|---|---|---|---|---|---|---|
| | 0.1 | | 0.2 | | 0.3 | | 0.4 | | 0.5 | |
| | mAP | rank-1 | mAP | rank-1 | mAP | rank-1 | mAP | rank-1 | mAP | rank-1 |
| BoT+EWTH | **87.7%** | 94.9% | **87.7%** | **95.0%** | 87.3% | 94.6% | 87.1% | 94.7% | 86.9% | **95.0%** |
| BoT+NEWTH | 88.3% | 95.0% | **88.4%** | 94.9% | 88.2% | 94.7% | 88.1% | **95.1%** | 88.0% | 95.0% |

Table 2: Results of series of EWTH loss in BoT with different values of $t$ on MSMT17

| Method | Values of $t$ | | | | | | | | | |
|---|---|---|---|---|---|---|---|---|---|---|
| | 0.1 | | 0.2 | | 0.3 | | 0.4 | | 0.5 | |
| | mAP | rank-1 | mAP | rank-1 | mAP | rank-1 | mAP | rank-1 | mAP | rank-1 |
| BoT+EWTH | 48.5% | **67.8%** | **48.7%** | 67.7% | 48.1% | 67.1% | 47.3% | 66.6% | 47.1% | 66.1% |
| BoT+NEWTH | **49.7%** | 67.8% | 49.5% | **68.1%** | **49.7%** | **68.1%** | 49.4% | 67.4% | 49.0% | 67.1% |

Table 3: Results of series of EWTH loss in AGW with different values of $t$ on Market1501

| Method | Values of $t$ | | | | | | | | | |
|---|---|---|---|---|---|---|---|---|---|---|
| | 0.1 | | 0.2 | | 0.3 | | 0.4 | | 0.5 | |
| | mAP | rank-1 | mAP | rank-1 | mAP | rank-1 | mAP | rank-1 | mAP | rank-1 |
| AGW+EWTH | **88.5%** | **95.4%** | 88.4% | 95.0% | **88.5%** | 95.3% | 88.0% | 95.3% | 87.8% | 95.2% |
| AGW+NEWTH | 89.0% | **95.6%** | 89.1% | 95.1% | **89.4%** | **95.6%** | 89.1% | 95.5% | 88.7% | 95.2% |

Table 4: Results of series of EWTH loss in AGW with different values of $t$ on MSMT17

| Method | Values of $t$ | | | | | | | | | |
|---|---|---|---|---|---|---|---|---|---|---|
| | 0.1 | | 0.2 | | 0.3 | | 0.4 | | 0.5 | |
| | mAP | rank-1 | mAP | rank-1 | mAP | rank-1 | mAP | rank-1 | mAP | rank-1 |
| AGW+EWTH | 49.9% | **69.7%** | 50.0% | 69.4% | 49.6% | 68.9% | **50.4%** | 69.4% | 49.1% | 68.5% |
| AGW+NEWTH | **53.1%** | **71.5%** | 51.8% | 70.1% | 52.8% | 71.0% | 52.2% | 70.6% | 50.4% | 69.7% |