[Reviews · NeurIPS 2020]

Review 1

Summary and Contributions: The paper proved that there is a dilemma in TriHard, where hard negative samples share certain characteristics in common with anchor samples and hard positive samples. But the elements of feature vectors extracted from these characteristics are forced to be away from each other by negative pairs but close to each other by positive pairs, which effect bad features clustering within classes. To mitigate TriHard loss dilemma, they propose several strategies and an Element-weighted TriHard loss. Extensive experiments are conducted on Market1501 and DukeMTMC-reID datasets and the achieve state-of-the-art results.

Strengths: 1. The paper proved the dilemma of TriHard loss from the perspective of gradient optimization. 2. The paper proposed targeted optimization based on the dilemma, such as half TriHard loss, TriHard loss with feature normalization, and half TriHard loss with average negative samples. Then element-weighted TriHard loss finally solved the dilemma.

Weaknesses: The paper makes element-weighted when training, but when testing, there is no weight to novel class. So the feature is only weighted when training. The inconsistency may limit accuracy.

Correctness: The claims and method correct are correct. The empirical methodology is correct, but with some weaknesses. See the comments in Weaknesses.

Clarity: The writing is well-understandable.

Relation to Prior Work: Yes

Reproducibility: Yes

Additional Feedback: after reading the rebuttal, I keep my original reate.


Review 2

Summary and Contributions: This paper aims to alleviate the dilemma of triplet loss when facing hard negative samples. The “dilemma” mentioned in this paper means that the similarity between the anchor and positive sample is useful for better representation, but the similarity between the anchor and negative sample should be repelled, so processing the similarity of anchor, positive sample and hard negative samples is a dilemma problem in triplet loss. To solve this problem, an Element-weighted TriHard Loss function is designed in this paper. The main idea is to weight the feature vectors of anchor and negative sample before calculating their distance to find discriminative elements of their feature vectors. Meanwhile, three mitigation strategies of TriHard loss dilemma are discussed in this paper. Experimental results on two datasets show that the proposed method can alleviate the dilemma by combining them with other strategies. The problems studied in this paper are meaningful for improving the stability of triplet loss. The discussed strategies and proposed method may bring some insights to the reader.

Strengths: + Both the motivation and proposed method of this paper are introduced clearly. + The discussion of several strategies that mitigate the influence of hard negative samples on triplet loss is helpful for other researchers. + The experiments of TH, TH+FH, HTH, HNTH and EWTH as well as their combinations will provide reference baselines for related research.

Weaknesses: - As shown in Table1, the proposed Element-weighted TriHard loss actually does not perform better than several existing strategies. Although combining EWTH with HTH or HNTH gets improved results, the effectiveness of the designed method of this paper is incremental. - the best value of t is varying for different methods and datasets (Supplementary materials), so it is too sensitive to selected. - Mathematical symbols in the Section of Method have some problem. For example, x_k under equation (5) has not been mentioned earlier. Because there are so many formulas, it's best to define all the symbols clearly in the paper.

Correctness: I think the proposed method and conducted experiments are correct and implementable.

Clarity: The paper is well written, but some typos should be corrected.

Relation to Prior Work: Yes, related work is well summarized.

Reproducibility: Yes

Additional Feedback: Post rebuttal: The authors address my concerns. I would like to keep my ratings.


Review 3

Summary and Contributions: This paper investigates the dilemma of TriHard loss, a problem that would cause unstable training. Through qualitative analysis, the authors believe this problem is caused by repelling the whole feature vector of anchor and hard samples when they share common elements. Hence, the authors introduce three simple strategies to alleviate this problem. They also propose an Element-weighted TriHard loss function. By putting the loss function and the strategies together, this paper achieves satisfying results.

Strengths: 1. The goal is apparent --- investigate and alleviate the dilemma of TriHard loss; 2. The performance is good --- compared with other loss functions; 3. There are alation studies; 4. The implementation code is attached, which is easy to follow.

Weaknesses: 1. There is no evidence provided to demonstrate the problem and some claims (such as L91-L93). 2. There is no training dynamics to identify why the proposed two techniques boost performance. 3. The proposed method is simple, like a combination of tricks. 4. Some parts of this paper are not clear, which makes reading difficult. I am not clear how do you plot figure 2 using D_{g1,g2}.

Correctness: I am not sure whether the claims and method are correct. Fisrt, some claims lack of evidences. Second, I am not carefully check the formulations and source code. Yes.

Clarity: No, some parts of this paper are not clear, which makes reading difficult.

Relation to Prior Work: No

Reproducibility: Yes

Additional Feedback: Post rebuttal: After reading the rebuttal and other reviews, I tend to change my rating to positive. I believe the authors can enhance this paper by providing more serious evidence to some claims and correcting the typos, as mentioned in the rebuttal. But considering its incremental novelty, my final rating is 6 -marginally above the acceptance threshold.


Review 4

Summary and Contributions: this paper analyses the weakness of triplet loss with batch hard theoretically and experimentally. Based on the analysis, the paper proposed a new method and proved to be very efficient on market and duke dataset.

Strengths: 1. the experimental results are very exciting 2. analysis is reasonable 2. code was provided as a material

Weaknesses: 1. writing need to be carefully polished 2. it will be better if some larger datasets are evluated, such as msmt17

Correctness: yes

Clarity: not very well

Relation to Prior Work: yes

Reproducibility: Yes

Additional Feedback: post rebuttal: after reading the rebuttal, I keep my original reate

[Author Response · NeurIPS 2020]

We appreciate the constructive comments and valuable points raised by the reviewers and the editor. We carefully read and discussed the opinions and doubts of each reviewer, and our responses are as follow.

**The writing problems.** The writing problems. Some expressions in this paper are not proper or brief enough. The writing needs to be more accurate, concise and academic. We will try our best to make the writing better.

**The consistency between training and inference.** Both outputs of training and inference are feature vectors of input images. The element weights are only used in the proposed element-weighted TriHard loss during training rather than components of the network. They make the loss function more accurate by focusing on different semantic parts among persons. It is an improvement on loss function without increasing the inference cost. Therefore, the consistency between training and inference is guaranteed.

**Doubts on Table 1.** Table 1 in our paper is the results of ablation study. These results fluctuate with or without some certain items of the proposed loss function to test the efficiency of each part. The analysis is detailed in chapter 5.1.

**Results fluctuate with parameter $t$.** Parameter $t$ is a hyper-parameter of element-weighted TriHard loss. Like the threshold in triplet loss, such hyper-parameters influence the performance of networks dramatically which is obtained by experience and experiments. That makes the training complex. It is a point to replace $t$ with a trainable parameter in our future work.

**Doubts on mathematical symbols.** We will add a parameter form into the next version of the paper.

**The evidence of some claims.** The claims in this paper without demonstration are proposed empirically. It will be more serious to cite previous work or attach experiment results to prove some of these claims. Such as the claim in L91-L93, we will attach an experiment result to it.

**Without training dynamics.** We are sorry to say that training dynamics figures are replaced by Figure 4, t-SNE visualization of feature distributions in our paper. Because of the limitation of pages and the effects they show, we made the decision.

**Doubts on Figure 2.** Figure 2 in our paper is plotted according to a classic visualization method CAM[1]. $D_{g1,g2}$ is not used.

**The proposed method is simple.** Our aim is to find a simple but serious method with good performance. Therefore, it is easy to follow and applied to other works.

**To evaluate the method on larger datasets MSMT17.** We replenish experiments on MSMT17[2] dataset, which contains more than 3 times identities than Market1501 and DukeMTMC-reID. The results are shown in Table 1.On AGW baseline, rank1 increases 3% and mAP 3.1% with $t = 0.3$. On BoT baseline, rank1 increases 4.7% and mAP 4.6% with $t = 0.3$. That proves the proposed loss improves the representational ability of the network to a great extent, which becomes more obvious on larger dataset.

Table 1: The performance of element-weighted TriHard loss on MSMT17

| Method | MSMT17 | | Method | MSMT17 | |
|---|---|---|---|---|---|
| | rank1 | mAP | | rank1 | mAP |
| BoT | 63.4% | 45.1% | BoT+HNEWTH | **68.1%** | **49.7%** |
| AGW | 68.2% | 49.5% | AGW+HNEWTH | **71.2%** | **52.6%** |

We demonstrate the importance of loss function in ReID networks. Only some simple improvements of TriHard loss will result in considerable enhancement of the performance. Without making any changes to the architecture of networks, it is easy to apply to other ReID or face recognition frameworks. The study of TriHard loss will also enlighten new researches on such commonly used loss functions in deep learning areas.

[1] B. Zhou, A. Khosla, A. Lapedriza, A. Oliva and A. Torralba, "Learning Deep Features for Discriminative Localization," 2016 IEEE Conference on Computer Vision and Pattern Recognition (CVPR) , 2016, pp. 2921-2929.

[2] L. Wei, S. Zhang, W. Gao and Q. Tian, "Person Transfer GAN to Bridge Domain Gap for Person Re-identification," 2018 IEEE/CVF Conference on Computer Vision and Pattern Recognition, 2018, pp. 79-88.


[Meta-Review · NeurIPS 2020]

All four knowledgeable reviewers were left with a favorable opinion of this work after the author rebuttal, and the AC agrees with this positive assessment. However, during the post-rebuttal discussion phase an independent ethics review was conducted regarding the general use of the DukeMTMC dataset. Some concerns raised about use of this dataset during this ethics review include: -- "... the dataset collection involved non-consensual video surveillance of students on Duke University campus. It is unlikely that all students even knew they were being recorded, and their relative lack of power with respect to the institution surveilling them also raises concerns about the ability to meaningfully object to the surveillance." -- "Including the dataset in the paper as-is would be problematic, as it would contribute to this mainstream use of the dataset. Referencing the issues and discouraging future use of the dataset would help mitigate this, as would full removal of the results." -- "The fact that others use the dataset uncritically does not make it appropriate, but it could have contributed to the authors being unaware of the issue, and that awareness may vary geographically." The recommendation of the NeurIPS Ethics Panel is that that *the DukeMTMC dataset should NOT be used in an accepted NeurIPS paper*. The AC therefore recommends a **conditional accept** (a rare and exceptional case reserved only for papers with ethical issues). This means the paper is accepted on the condition that the final version 1) removes all results on DukeMTMC, 2) makes it clear that the dataset has been taken down and should no longer be used, and 3) reproduces all technical contributions and results of the original submission using other datasets. Moreover, the AC strongly recommends that the authors use the Broader Impact section to state clearly that surveillance is the typical goal of re-id systems, that re-id systems often rely on non-consensual surveillance data for their training, and to discuss and generally raise awareness about the consent challenges with MTMC data collection. The authors should also address the suggestions of R3 regarding better evidence in support of key claims and improvements to the writing. The AC discussed this decision with the SAC and PCs. ******************************* Note from Program Chairs: The camera-ready version of this paper has been reviewed with regard to the conditions listed above, and this paper is now fully accepted for publication.